# Correction: Cecchi et al. Perioperative Treatments in Pleural Mesothelioma: State of the Art and Future Directions. *Cancers* 2025, *17*, 3199

**DOI:** 10.3390/cancers18010103

**Published:** 2025-12-29

**Authors:** Luigi Giovanni Cecchi, Marta Aliprandi, Fabio De Vincenzo, Matteo Perrino, Nadia Cordua, Federica Borea, Alessandro Bertocchi, Antonio Federico, Giuseppe Marulli, Armando Santoro, Giovanni Luca Ceresoli, Paolo Andrea Zucali

**Affiliations:** 1Department of Biomedical Sciences, Humanitas University, Via Rita Levi Montalcini 4, Pieve Emanuele, 20072 Milan, Italy; luigi.cecchi@humanitas.it (L.G.C.); alessandrp.bertocchi@cancercenter.humanitas.it (A.B.); antonio.federico@cancercenter.humanitas.it (A.F.); giuseppe.marulli@hunimed.eu (G.M.); armando.santoro@cancercenter.humanitas.it (A.S.); paolo.zucali@hunimed.eu (P.A.Z.); 2Department of Oncology IRCCS, Humanitas Research Hospital, Via Manzoni 56, Rozzano, 20089 Milan, Italy; fabio.de_vincenzo@cancercenter.humanitas.it (F.D.V.); matteo.perrino@humanitas.it (M.P.); nadia.cordua@cancercenter.humanitas.it (N.C.); 3Department of Oncology, Cernusco sul Naviglio, ASST Melegnano e Martesana, 20063 Milan, Italy; federica.borea@asst-melegnano-martesana.it; 4Division of Thoracic Surgery, IRCCS Humanitas Research Hospital, Via Manzoni 56, Rozzano, 20089 Milan, Italy; 5Department of Medical Oncology, Humanitas Gavazzeni, 24125 Bergamo, Italy

## Text Correction

There was an error in the original publication [1]. The following sentence is missing in the Section 6 Dicussion, Paragraph 4: It is notable that, among the surgical community, there is a general agreement that P/D should be privileged over EPP when technically feasible, as results are comparable in term of DFS and OS but with significantly lower morbidity and mortality, thus leaving more chance for adjuvant treatments and improving quality of life [102].

A correction has been made to Section 6 Discussion, Paragraph 4.

Considering the results of the EORTC 1205 study, the decision on each individual case should be based on the careful evaluation of various factors, including the disease extension, the type of surgical intervention, and the risk factors for postoperative morbidity. It is notable that, among the surgical community, there is a general agreement that P/D should be privileged over EPP when technically feasible, as results are comparable in term of DFS and OS but with significantly lower morbidity and mortality, thus leaving more chance for adjuvant treatments and improving quality of life [102].

## Reference Correction

In the original publication, there was an error in the Reference list. The reference “Marulli, G.; Breda, C.; Ratto, G.B.; Leoncini, G.; Alloisio, M.; Infante, M.; Luzzi, L.; Paladini, P.; Oliaro, A.; Ruffini, E.; et al. Pleurectomy-decortication in malignant pleural mesothelioma: Are different surgical techniques associated with different outcomes? Results from a multicentre study. *Eur. J. Cardiothorac. Surg.* **2017**, *52*, 63–69. https://doi.org/10.1093/ejcts/ezx079” was not included in the reference list. The correct reference number is [102]. 

The authors state that the scientific conclusions are unaffected. This correction was approved by the Academic Editor. The original publication has also been updated.

## References

[B1-cancers-18-00103] Cecchi L.G., Aliprandi M., De Vincenzo F., Perrino M., Cordua N., Borea F., Bertocchi A., Federico A., Marulli G., Santoro A. (2025). Perioperative Treatments in Pleural Mesothelioma: State of the Art and Future Directions. Cancers.

