# Peer review of "Correction: Cecchi et al. Perioperative Treatments in Pleural Mesothelioma: State of the Art and Future Directions. Cancers 2025, 17, 3199"

_cancers, 2025, doi:10.3390/cancers18010103_

Round 1

Reviewer 1 Report

Comments and Suggestions for Authors

I would like to thank the authors for their submission. The review article “Perioperative treatments in pleural mesothelioma: state of the art and future directions” addresses an important topic in thoracic oncology, providing a comprehensive overview that is potentially valuable for both clinical practice and future research. Overall, the article is well-structured and clearly presented.

To further strengthen the manuscript, I would suggest that the listed studies be accompanied by their year of publication, which would provide readers with a clearer chronological framework. In addition, while the overview is very strong, a somewhat greater focus on intrathoracic therapies would enrich the discussion. This area is particularly exciting and rapidly evolving, with approaches such as CAR-T cell therapy and several additional pro- and retrospective studies accumulating extensive experience with HITHOC beyond those already mentioned in the manuscript.

I would also like to point out that the clinical trial number NCT02040272, as currently cited in the manuscript, is incorrect. This identifier actually refers to the MARS 2 trial led by Professor Eric Lim and is not related to HITHOC. I would kindly suggest correcting this trial number in the manuscript. Additionally, it would be helpful to double-check all other trial identifiers listed in the tables to ensure that each corresponds accurately to the intended study. This will improve clarity and reliability for readers referencing these studies.

Overall, I consider my comments to be minor revisions, as the manuscript provides an excellent and timely overview.

I believe the manuscript will be suitable for publication after these minor revisions. The article represents a valuable and timely contribution to the field of thoracic oncology.

Author Response

Comment 1: 

To further strengthen the manuscript, I would suggest that the listed studies be accompanied by their year of publication, which would provide readers with a clearer chronological framework.

Reply to comment 1: 

Thank you for this valuable suggestion. We agree that including the year of publication will provide readers with a clearer chronological framework of the available evidence. We have therefore added the publication years to the studies listed in Tables 1 and 2.

Comment 2: 

In addition, while the overview is very strong, a somewhat greater focus on intrathoracic therapies would enrich the discussion. This area is particularly exciting and rapidly evolving, with approaches such as CAR-T cell therapy and several additional pro- and retrospective studies accumulating extensive experience with HITHOC beyond those already mentioned in the manuscript.

Reply comment 2:

Thank you for this insightful comment. We have expanded the discussion by adding recent publications and evidence on HITHOC in the context of MPM, including the results of Elsayed et al. (Updates in Surgery). These additions can be found on page 9, sentences 339–355.

Regarding CAR-T cell therapy, we have included a section in the immunotherapy paragraph (pages 11–12, sentences 459–475) summarizing the available data. To our knowledge, however, there are currently no ongoing clinical trials listed on clinicaltrials.gov evaluating their use in the perioperative setting.

Comment 3: 

I would also like to point out that the clinical trial number NCT02040272, as currently cited in the manuscript, is incorrect. This identifier actually refers to the MARS 2 trial led by Professor Eric Lim and is not related to HITHOC. I would kindly suggest correcting this trial number in the manuscript. Additionally, it would be helpful to double-check all other trial identifiers listed in the tables to ensure that each corresponds accurately to the intended study. This will improve clarity and reliability for readers referencing these studies.

Reply to comment 3: 

Thank you for pointing this out — and our apologies for the error. We have corrected the mistaken trial identifier and updated the corresponding reference. In addition, we have carefully double-checked all clinical trial identification codes and their references throughout the manuscript (tables and main text) and corrected any inconsistencies to ensure accuracy and clarity.

Reviewer 2 Report

Comments and Suggestions for Authors

Dear Editor and Authors,

Thank you for asking me to review this work titled "Perioperative treatments in pleural mesothelioma: state of the art and future directions" by Dr. Luigi Giovanni Cecchi and colleagues from Milan, Italy.

In this work the authors present a quite comprehensive and detailed overview of perioperative oncological treatment of malignant pleural mesothelioma. The manuscript is well written and organized with each modality receiving a thorough overview and literature review!! The information provided are modern and up to date and the organization and presentation of the data is quite good. Consequently this work is a benefit to the reader who gains important insight on different therapeutic modalites. I only have some very minor issues with the work following a extensive review of it!!

Comments:

  1. As a thoracic surgeon I dissagree with the "quantifier" merley in line 80!! How many decortications do the authors feel are adequate for an experienced thoracic surgeon to obtain competency in pleurectomy??
  2. One of the observations this reviewe made is that although the review is surgically oriented, not a single thoracic surgeon has contributed to the manuscript!! This alone is quite a significant limitation as the authors who are oncologist do not posess the surgical insight of an experienced surgeon!!!  

In conclusion, I feel following a very minor editing the work can be presented / published!! Thank you. 

Author Response

Comment 1 : 

As a thoracic surgeon I dissagree with the "quantifier" merley in line 80!! How many decortications do the authors feel are adequate for an experienced thoracic surgeon to obtain competency in pleurectomy??

Reply to comment 1:

Thank you for your observation and thoughtful comment. We agree that it is difficult to dichotomize centers or surgeons solely based on the total number of pleurectomies performed, and that a single numerical “quantifier” may not adequately reflect surgical expertise.

Several studies across different cancer types, including MPM, have shown that treatment in high-volume centers is associated with improved outcomes, such as shorter postoperative hospitalization (p = 0.035), lower 30-day readmission rates (4.6% vs. 6.1%, p = 0.021), and lower 90-day mortality (10.0% vs. 14.6%, p = 0.029), as reported by Verma et al. (Lung Cancer, 2018). In that analysis, high-volume centers were defined as those above the 90th percentile of case volume.

In the MARS2 trial, UK centers were required to meet specific quality-assurance criteria, including at least two mesothelioma surgeons who had undergone peer-reviewed observation of (extended) pleurectomy/decortication, completion of a minimum of five procedures, and observation of the first trial case by a surgeon from the pilot phase. However, these requirements do not necessarily align with the 90th-percentile threshold reported in registry analyses.

We agree that even the 90th-percentile cutoff may not represent the optimal definition of a high-volume center or of individual surgical competency. This remains an unmet need that should be addressed in future research. Moreover, variability in surgeon and center experience within the MARS2 trial may represent a potential confounding factor influencing outcomes.

Comment 2: 

One of the observations this reviewe made is that although the review is surgically oriented, not a single thoracic surgeon has contributed to the manuscript!! This alone is quite a significant limitation as the authors who are oncologist do not posess the surgical insight of an experienced surgeon!!!  

Reply to comment 2: 

Thank you very much for this insightful observation. We completely agree that surgical expertise is essential in the management of malignant pleural mesothelioma, and we appreciate you drawing our attention to this important point.

To strengthen the manuscript, we involved Professor Giuseppe Marulli, Head of Thoracic Surgery at our institution, who kindly reviewed the work and has now been included as a co-author.